# Relationship between Children’s and Parents’ Dental Anxiety: A Cross-Sectional Study on the Six European Countries

**DOI:** 10.3390/dj10110209

**Published:** 2022-11-04

**Authors:** Luka Šimunović, Bruno Špiljak, Milica Radulović, Adna Vlahovljak, Mihailo Ostojić, Jovan Krlev, Amina Ibrahimpašić, Lara Vranić, Dubravka Negovetić Vranić

**Affiliations:** 1School of Dental Medicine Zagreb, University of Zagreb, 10000 Zagreb, Croatia; 2Department of Dental Morphology and Gnathology, Dental Medicine Studies, Medical Faculty in Podgorica, University of Montenegro, 8290 Podgorica, Montenegro; 3Faculty of Dental Medicine, University of Belgrade, 11000 Beograd, Serbia; 4School of Dentistry Skopje, 1000 Skopje, North Macedonia; 5Faculty of Medicine, University of Ljubljana, 1000 Ljubljana, Slovenia; 6Department of Pediatric and Preventive Dentistry, School of Dental Medicine, University of Zagreb, 10000 Zagreb, Croatia

**Keywords:** dental anxiety, parents, children, Europe, psychology

## Abstract

Background: The purpose of this study is to investigate the relationship between children’s and parents’ dental anxiety. Methods: 731 children of different ages and their parents from six European countries participated in this study. Dental anxiety was investigated through an online questionnaire, which consisted of general questions and the Corah Dental Anxiety Scale (CDAS), which is a questionnaire that measures respondents’ reactions on a 5-point scale for four different situations. Results: CDAS results were calculated for all children and their parents. A total of 12.5% of children from Croatia, 26.67% from Macedonia, 10.94% from Bosnia and Herzegovina, 20.31% from Montenegro, 23.08% from Slovenia and 16.10% from Serbia showed a high level of anxiety. The correlation between dental anxiety of parents and children was 0.4 (*p* < 0.01). Conclusions: Parents with negative experience from a dental office can have a bad effect on their child’s behaviour, which results in the creation of a non-active patient. Due to the clear and existing cause-and-effect relationship of dental anxiety in children and parents, it is extremely important to educate parents about the proper psychological approach to children in order to promote positive experiences from dental offices, as well as to emphasize the importance of regular visits to the dentist.

## 1. Introduction

Dental fear and anxiety (DFA) are problems that dentists encounter on a daily basis when working with both children and adult patients. With a prevalence of 9% in pediatric dentistry, they pose a major challenge for any physician who encounters patients of that age [1,2]. In adult patients, they are most often a reflection of previous negative experiences from dental practices that are associated with childhood and adolescence. Dental fear is an immediate and disturbing reaction that arises in response to an individual’s reaction to threatening situations or events [3,4]. Additionally, a previous negative experience in the dental office may be a cause of fear at the next checkup. Dental anxiety (DA) is stronger than fear in its intensity and is irrational in relation to fear [5]. It is defined as a non-specific feeling of anxiety that arises in response to possible threats or dangers that occur in stressful situations where the source of the threat is unclear. This emotional reaction does not necessarily have to be triggered by an object, but the very thought of one of the dental stimuli can be the trigger for an anxiety reaction [1]. Therefore, dental anxiety can also be defined as a general, non-specific dislike of dental care, the doctor who performs it, or the dental practice [6] In his critical examination, Rachman identified three possible triggers for anxiety: his own past negative experience, imitation of someone else’s anxiety, and verbal introduction to dental treatment [7]. Other studies consider Rachman’s third trigger for anxiety to be just the opposite, i.e., a way to calm the patient because the reaction to the conversation depends on the type of frightened patient [8]. The consequences of dental anxiety are delaying or avoiding a visit to the dentist, difficulties in treating such patients, increased incidence of stress when visiting the dentist, and poorer oral health. If anxiety is not diagnosed in time, a “vicious circle of dental anxiety” occurs. Dental fear and anxiety lead to the neglect of oral health, resulting in a worsening of the problem, and this leads to longer and more painful treatment that increases fear in the patient and leads to re-avoidance of the office and the entire dental team [9,10]. In a study by Mueller et al. [11] from 2022, dental anxiety and negative dental experiences significantly reduced the participants’ perception of dental self-efficacy, as well as dental self-examination. Patients with dental fear and anxiety can sometimes make the procedure extremely stressful for dental staff, which, in turn, can disrupt interpersonal relationships in the office, as well as the relationship between the patients, parents and doctors, which is a key factor in solving DFA problems and shaping behaviour in the dental office [1,12,13]. Communication between the three components of this triangle (the interdependence of each component on the attitude of the other two) is a key factor in creating the conditions appropriate for performing dental treatment [14]. It has been discovered and generally accepted that the genesis of dental anxiety occurs in childhood and is not inherited. Reasonable speculation is that these early dental fears shape the patient’s attitude in adulthood. Research has shown that adults who have negative attitudes about going to the dentist can and do transmit such attitudes to their children [15]. This results in the avoidance of dental care and the lack of regular visits to the dentist. Therefore, it is clear that negative attitudes tend to continue. A review by Hegde et al. from 2022 [16] pointed out that parental DFA influences the behavior of adolescents and, therefore, can influence the seeking of dental care. For this reason, it is extremely important to pay attention to parental DFA before their child’s intervention and treatment. Comments from parents and peers about dental treatment strongly influence the fear of dental treatment [1,17]. Some studies have shown a close link between maternal and child dental anxiety. Maternal anxiety before the child’s dental procedure was found to be significantly associated with the child’s dental phobia [1,17,18,19]. However, the family environment has changed with the increasing number of single parents, mixed families, etc. Mothers do not always bring their children to the dentist—sometimes fathers, both parents or caregivers do so. This leads to the conclusion that the connection between the anxiety of parents and children should not be investigated exclusively with mothers. A 2015 study by Khawja and colleagues found that parental dental anxiety directly affects a child’s dental anxiety and found a significant association between maternal dental anxiety and increased child caries findings [20]. When assessing children’s behavior, it is very important to pay attention to the correct choice of method based on the psychological development of the child and the degree of dental anxiety, the correct way of conducting measurements and the correct interpretation of the results. One of the best-known measurement scales used to assess dental anxiety is the Corah Dental Anxiety Scale (CDAS or DAS). The test is very simple, has a high coefficient of reliability, and it takes about five minutes to complete. These are the main reasons for its most common application in everyday clinical practice [20,21,22,23]. When reviewing the literature on dental anxiety, it became apparent that much of the background research had been done in nations with vastly different cultural norms and healthcare systems [24,25,26,27]. Therefore, it is crucial to take into account any potential parallels and discrepancies between the background research from other nations and the results of the current study. With regard to the aforementioned, the reason for comparing these countries lies in the fact that these countries of southeastern and Central Europe were part of the same entity (former Yugoslavia) until 1991. The aim of this study was to investigate the relationship between children’s and parental dental anxiety in those six countries.

## 2. Materials and Methods

### 2.1. Study Design

The survey was conducted between December 2020 and March 2021. A total of 731 children and their parents from selected countries participated in the research—Croatia (*n* = 120), Macedonia (*n* = 120), Bosnia and Herzegovina (*n* = 128), Montenegro (*n* = 128), Slovenia (*n* = 117) and Serbia (*n* = 118). The reason for comparing these countries lies in the fact that these countries of southeastern and Central Europe were part of the same entity until 1991 so they share common cultural origin [28]. Total sample size required determining whether a correlation coefficient differs from zero for: α (two-tailed) = 0.05; β = 0.01; γ = 0.38 (exp) equals 117. Participation was voluntary and anonymous. Dental anxiety was investigated through an online questionnaire on children of primary, secondary and university age and their parents. The age of the children ranged from 5 to 28 years. 207 respondents attended primary school, 285 secondary school, 35 polytechnic school and 204 university. To access the study itself, respondents had to read and sign informed consent, after which they were granted access to the survey itself. The survey was anonymous and approved by the Ethics Committee of the School of Dental medicine Zagreb (number: 05-PA-30-XXVIII-6/2021) in accordance with the ethical standards of the Declaration of Helsinki.

### 2.2. Questioner Design

The survey questionnaire consisted of 2 parts: general questions and the Corah Dental Anxiety Scale (CDAS) questionnaire (Figure A1). Through general questions, information was obtained on the age of the respondents and the level of education while the CDAS measured the reactions of the respondents on a 5-point scale for 4 different situations.

### 2.3. Statistical Analysis

Descriptive statistics were processed by the program Statistica (TIBCO^®^ Statistica™ Version 13.5.0.17., Palo Alto, CA, USA). The results were compared using the Kruskal–Wallis test and the post hoc Dunn test. Correlation coefficients were calculated by the Spearman rank test.

## 3. Results

CDAS results were calculated for all children and their parents. Based on CDAS results, all subjects were divided into three groups: low (CDAS 4–8), moderate (CDAS 9–12) and high anxiety (CDAS 13–20) (Figure 1). The largest share of parents with a high degree of dental anxiety was in Serbia (18.64%), while the largest share of children was in Macedonia (26.67%). Children (mean 8.7264) showed more anxiety than their parents (mean 8.2832). In terms of the level of children’s education, 29.95% of primary school children, 12.28% of secondary school children, 25.71% of children attending college or university and 13.24% of children in college showed a high level of anxiety. The geographical presentation of high levels of dental anxiety is shown in Figure 2.

Kruskal–Wallis test H (5, *n* = 731) = 16.10819 *p* = 0.0065 confirmed a statistically significant difference between the surveyed countries in the category of parental dental anxiety and rejected the null hypothesis. The most significant difference was found between the dental anxiety of parents from Bosnia and Herzegovina, compared to the dental anxiety of parents from Slovenia (*p* = 0.0039), while the comparison of children’s anxiety by country is shown in Table 1. The level of anxiety of children from Slovenia was statistically significantly higher than the level of children’s anxiety in Serbia (*p* = 0.0037), Croatia (*p* = 0.0206) and Bosnia and Herzegovina (*p* = 0.0315). Spearman’s rank test showed the correlation between dental anxiety of children and parents by individual countries (Table 2); dental anxiety of parents and their children depending on age, i.e., the level of education the child is currently attending (Table 3); and dental anxiety of parents and children attending primary school and high school under 18 (Table 4). The correlation between dental anxiety of all examined parents and their children was 0.4063 (*p* < 0.01).

## 4. Discussion

A family is crucial for the upbringing and development of a child. It is the family that gives the child a sense of socialization, care, love and appreciation. Not only dentists but also parents are involved in promoting oral health. They can significantly influence the child’s behavior during a visit to a dental office. The family influences the child’s oral health in everyday life by taking care of the diet and encouraging the practice of oral hygiene. Parents are the ones who decide when their child will visit the dentist, they choose the clinic and the doctor, organize the visit and accompany the child to the office. Before coming to the office, the parents have to prepare the child for treatment by following the advice of the dentist. In this way—with parental preparation at home and the application of methods of shaping behavior in the office—the child is gradually prepared for any planned procedure without the development of dental fear and anxiety [10,35]. With this research, we have shown that dental anxiety is high in the area of southeast Europe. In some countries, as many as a quarter of children show a high degree of anxiety (Figure 1). It has been proven that a child’s behavior is the result of genetic predisposition (genes and interactions between genetic potentials and the environment) and upbringing. Some patients have developed anxiety without having been previously exposed to any negative experience or information. In such patients, the development of dental fear and anxiety is influenced by endogenous factors, such as genetic vulnerability, personality traits, age and gender [36]. In other patients, anxiety developed due to exogenous factors as a result of direct or indirect conditioning. With immediate conditioning, anxiety develops as a result of one’s own negative experience from the office. Indirect conditioning arises as a result of observing another person’s anxious behavior and imitating that behavior (modelling) or is the result of exposure to negative information, most often obtained from parents, family members, peers, teachers, television or social media. Research has shown that immediate conditioning had a greater impact on the development of DFA compared to the other two modes of indirect conditioning [7]. Parenting styles were thought to shape children’s behaviors by providing an environmental framework for their psychosocial development [37]. Baumrind identified three parenting styles: authoritative, authoritarian and permissive [38]. This classification of parenting styles has proven to be a useful tool for researching the impact of parenting on various aspects of child development [38]. Scholarly interest has been drawn to the relationship between parenting styles and children’s dental fear. A study found that the subscale self-complaints (example item “My child’s happiness requires a lot of sacrifice on my part”) were associated with children’s dental fear [39]. Another study, however, found no link between parenting styles and children’s dental anxiety [40]. The highest level of dental anxiety in children in this study was observed in Macedonia and Slovenia, which does not correlate with the available Decayed, Missing and Filled Teeth (DMFT) index data in these countries: in Macedonia DMFT index was 3.5 for 12-year-olds and in Slovenia DMFT index was 1.5 [29,30]. This confirms that high DMFT index and “bad” experience with dentists are not the only factors that contribute to a high level of anxiety. In our study, the high degree of parental anxiety varied from 5–19% within the countries surveyed (Figure 1). Today, it is estimated that 11–20% of the adult population experiences severe dental anxiety. Children’s dental anxiety is associated with their parents’ anxiety. Negative attitudes towards the practice and family dentists are common reasons for the development of dental anxiety among children. When it comes to the occurrence of dental anxiety, numerous studies have shown that there is a connection between parents and children [41]. A parent–child anxiety correlation of 0.4 in our study suggests a strong association (Table 2). Bad experiences of parents and the negative connotation of going to the dentist, which parents unconsciously show and comment on in front of children, builds an image of the dentist as a person who one “should” fear. Some frightened parents testify that their anxiety began in childhood, in some cases even before the first visit to the dentist [42]. It is known that the level of anxiety in children is proportional to the number of people, and especially family members who are afraid of dentists [43]. A 2012 study refuted the theory that dental anxiety is transmitted mostly through the mother, as previously thought. Namely, it turned out that the fathers are also intermediaries in the transmission of information about danger and unpleasant situations, including dental anxiety [17]. Additionally, in the past, children were more regularly brought to the dental office by their mothers, and today mothers and fathers do the same. If we consider only primary and secondary school children, the correlation of anxiety between parents and children becomes even more significant (0.48). Younger children have been shown to perceive maternal anxiety more subtly, with levels of dental anxiety significantly related to maternal anxiety levels [44], while older children show a high degree of maturation and independence, so they are less affected by maternal anxiety [45]. In a study by Škrinjarić et al. on the correlation between the DMFT index and the anxiety of children at the Institute of Pedodontics in Zagreb (children in primary and secondary schools) and their mothers, according to the CDAS scale, a correlation of 0.38 was obtained, which agrees with our research (the correlation of children under 18 years of age and their parents in the Republic of Croatia is 0.4) [46]. If we compare the DMFT index in Macedonia and Croatia, whose data are available to us from 2013, they are relatively similar (3.5 and 4.2) [29,31]. However, if we take into account the high level of dental anxiety shown in this study, we can notice that there is a big difference. In Croatia it is 12.5%, while in Macedonia it is 26.67%. This could be explained with the successfully introduced national program in the Republic of Croatia, which has been implemented since 2017 under the name of “Zubna putovnica” (translated as “Dental Passport”) [47]. The program is aimed at improving the oral health and health behavior of school and preschool children with the aim of early detection and prevention of caries. The national program began in the 2017/2018 school year. Sixth graders (12 years old) and preschoolers entering the first grade (6 years old) received the forms directly from the school medicine doctor or at school. It became the part of the mandatory medical documentation collected by the school doctor during the systematic examination when enrolling in the first grade of elementary school. The form was collected from 6th grade students during the following regular activities: Hepatitis B vaccination, spinal examination, and height and weight control (growth and development monitoring). The completed form was returned to the school medicine doctor by the child/parent/guardian following the dental examination. It includes the child’s first and last name, gender, birth year, dental status for primary and permanent teeth, marks for tooth status (for calculating the DMFT index) and information on preventive and therapeutic procedures performed at that visit or scheduled for the next visit. According to Central Information Health System of the Republic of Croatia (CEZIH) data, from 1 September to 31 December 2017, there was an increase in the number of first examinations and diagnostic-therapeutic procedures in children aged 12 years, compared to the same period in 2015. Fissure sealing and sealing restorations were the least commonly reported preventive procedures in preschool children and sixth grade students, while motivating and instructing children on oral hygiene were the most commonly reported procedures [47]. All of the above can be summarized and explained from a psychological point of view. Today, we continue to adopt Piaget’s understanding of infants and young children as active, curious, and engaged students, who build knowledge by generating, testing, and developing theories to explain their world. Understanding children in this way means that we should present environments that intrigue them and provide them with opportunities to participate in trial and error. Even small changes in our approach to children can make this possible. “Egocentrism” in adolescence involving an imaginary audience can simply be characterized: “In the mind of a young person, he is always on the stage” [48]. Given the feeling that they are “on stage” and that everyone is watching them, adolescents can spend time imagining how they will be perceived. With this psychological explanation, it is clear to us why there is a higher rate of highly anxious children in primary and secondary schools, compared to those in college. One of the possible ways to reduce the anxiety transmission from parent to child can be achieved by educating parents about the psychological approach and preparing the child for going to the dental office but also by raising awareness about the importance of oral hygiene, as well as prophylactic and preventive procedures. Considering the overload of the public health system, which is primarily manifested in overcrowded waiting rooms of dental medicine offices, as well as the breakthrough of the infection of COVID-19, the development of telemedicine and teledentistry could be crucial for relieving the burden on the public health system. It is a field of healthcare that uses technological assets for the purpose of exchanging clinical information and images “remotely”, enabling consultation with a doctor/dentist, diagnosis and therapy planning [49,50,51,52]. The available literature states that there are three basic forms of telestomatology: data storage and forwarding (asynchronous), dentist–patient interaction in real time (synchronous) and management and monitoring of dental status and/or promotion of prophylactic and preventive measures through mobile technology (so-called mobile health services). Teleconsultations are the most common form of telestomatology. They are of particular importance considering that they reduce the number of non-urgent arrivals of patients in dental offices, as well as the number of referrals from primary health centers to higher centers, and thus relieve the public health system [53,54]. Teleconsultations can be done through instant messaging applications or video calls. In addition to teleconsultations, other forms of telestomatology include telediagnosis, teletriage and telemonitoring. Telediagnostics enables the exchange of extraoral, intraoral and radiographic images, on the basis of which a diagnosis of certain oral pathological changes can be made [55,56]. Teletriage serves as a supplement or replacement of the clinician’s interaction with the patient, it includes the correct, safe and timely collection of symptoms and signs reported by the patient via smartphone, it enables the examination of the patient, as well as the determination of his condition and then offers the necessary care. One of the latest technological achievements, telemonitoring, is aimed at replacing face-to-face visits with virtual visits for the purpose of regular monitoring of treatment outcomes, as well as the progression of the disease itself [53]. Available data in the literature suggest that telestomatology can be used to support the conventional approach to the treatment of various oral diseases, such as temporomandibular disorders [57]. For example, patients suffering from oral lichen planus can be educated about the rule of titration of corticosteroid therapy when the disease is in an active phase and then monitored via teleconsultations [58]. However, despite being a starting point for providing higher quality information, doctors and dentists need to be made aware that patients can receive various wrong and biased information through digital platforms, which is why additional work is needed on educating the general population, as well as health professionals [51]. Taking into account all of the above, it is necessary to consider the application of telestomatology for the purpose of educating parents about the approach and importance of the perception of dental care and oral health. It is also necessary to make parents aware of the consequences of non-compliance and their impact on the quality of life of the child [59]. Furthermore, because most modern dental procedures are not inherently painful or as anxiety-provoking as they once were, future research should investigate the cause of the persistence of dental anxiety over time and work to reduce the prevalence of dental anxiety.

## 5. Conclusions

In the dental practice, patients with dental fear and anxiety, whether children or adults, are an increasingly common problem. Dental anxiety develops very early, often before the first visit to the dentist, in which parents play an important role. With their own personal negative experience, they can have a negative effect on their child’s behavior, and thus create a non-cooperative patient. Both father and mother are mediators in the transmission of negative experiences and information, as well as anxiety itself, and the sociodemographic, behavioral and psychosocial characteristics of parents and immediate families are closely related to the development of dental anxiety. In this study, the correlation between dental anxiety of parents and children was statistically significant. With the development of dental care, there has been a reduction in the discomfort and pain of dental procedures, so future studies should be directed towards the reasons for the persistent high levels of anxiety, both in parents and children. Furthermore, it is necessary to consider the possibilities of telestomatology as an extension of the existing dental system, but also as a tool for psychological preparation and education of parents. Moreover, parents can greatly influence their child’s behavior, so educating parents before their child goes to the dentist can be helpful in promoting a positive experience from the dental office.

## Figures and Tables

**Figure 1 dentistry-10-00209-f001:**
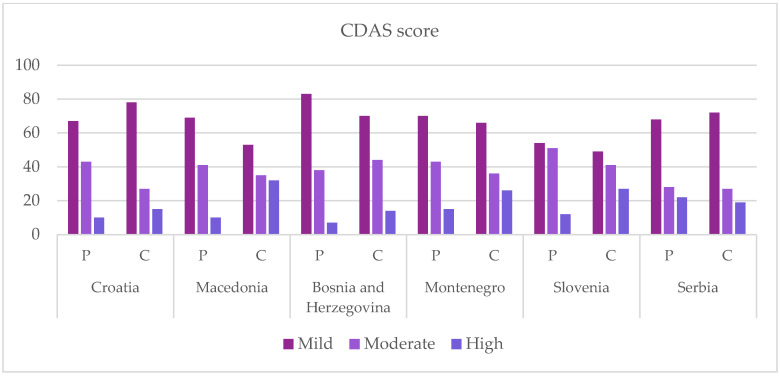
CDAS (Corah Dental Anxiety Scale) distribution: low (CDAS results 4–8), moderate (CDAS results 9–12) and high anxiety (CDAS results 13–20); P—parents; C—children.

**Figure 2 dentistry-10-00209-f002:**
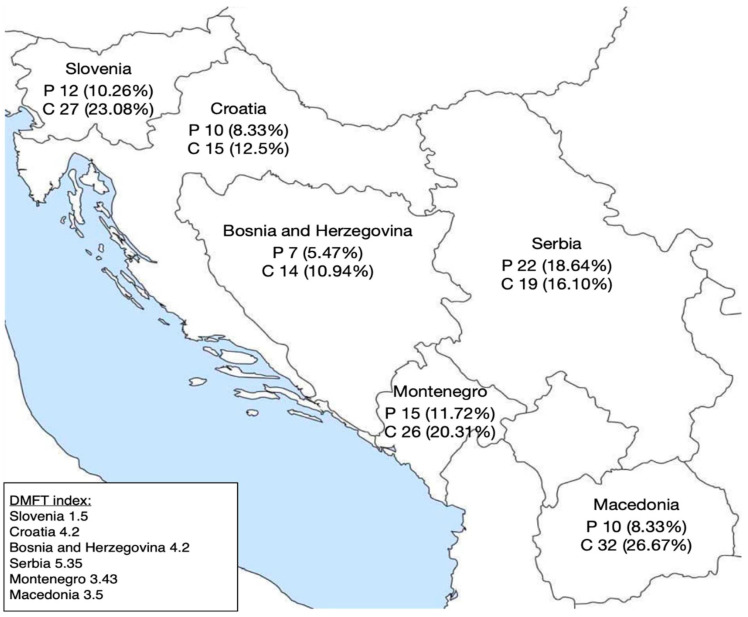
The geographical presentation of dental anxiety high levels and World Health Organization’s Decayed, Missing and Filled Teeth (DMFT) index [29,30,31,32,33,34]. [P—parent; C—child].

**Table 1 dentistry-10-00209-t001:** Comparison of children’s dental anxiety (CDAS) by country; multiple comparisons *p* values (2-tailed); children dental anxiety (DA); independent (grouping) variable: country; Kruskal–Wallis test: H (5, *n* = 731) = 23.40126 *p* = 0.0003.

	Croatia *R:332.85*	Macedonia *R:401.00*	BiH *R:337.58*	Montenegro *R:385.34*	Slovenia *R:420.67*	Serbia *R:319.76*
Croatia		0.186357	1.000000	0.757090	0.020566 *	1.000000
Macedonia	0.186357		0.271423	1.000000	1.000000	0.045006 *
BiH	1.000000	0.271423		1.000000	0.031451 *	1.000000
Montenegro	0.757090	1.000000	1.000000		1.000000	0.224381
Slovenia	0.020566 *	1.000000	0.031451 *	1.000000		0.003743 *
Serbia	1.000000	0.045006 *	1.000000	0.224381	0.003743 *	

BiH—Bosnia and Herzegovina; * statistical significance (*p* < 0.05)

**Table 2 dentistry-10-00209-t002:** Correlation between dental anxiety of children and parents by individual countries; Spearman rank correlation (marked correlations are significant at *p* < 0.01).

CDAS	Croatia		Macedonia		BiH		Montenegro		Slovenia		Serbia		All	
	DA (P)	DA (C)	DA (P)	DA (C)	DA (P)	DA (C)	DA (P)	DA (C)	DA (P)	DA (C)	DA (P)	DA (C)	DA (P)	DA (C)
DA parents	1.000000	0.171952	1.000000	0.616837 *	1.000000	0.668555 *	1.000000	0.280708 *	1.000000	0.475016 *	1.000000	0.220034	1.000000	0.406368 *
DA children	0.171952	1.000000	0.616837 *	1.000000	0.668555 *	1.000000	0.280708 *	1.000000	0.475016 *	1.000000	0.220034	1.000000	0.406368 *	1.000000

DA—dental anxiety; P—parent; C—child; BiH—Bosnia and Herzegovina; CDAS—Corah dental anxiety scale; * statistical significance (*p* < 0.01)

**Table 3 dentistry-10-00209-t003:** Correlation between dental anxiety of parents and their children depending on the level of education the child is currently attending; Spearman rank correlation (marked correlations are significant at *p* < 0.01).

CDAS	Primary School		Secondary School		Polytechnic School		College or University	
	DA (P)	DA (C)	DA (P)	DA (C)	DA (P)	DA (C)	DA (P)	DA (P)
Parent’s DA	1.000000	0.395822 *	1.000000	0.566812 *	1.000000	0.386092	1.000000	0.255956 *
Child’s DA	0.395822 *	1.000000	0.566812 *	1.000000	0.386092	1.000000	0.255956 *	1.000000

DA—dental anxiety; P—parent; C—child; CDAS—Corah dental anxiety scale. * statistical significance (*p* < 0.01)

**Table 4 dentistry-10-00209-t004:** Correlation between dental anxiety of parents and children attending primary school and high school under 18 years of age; Spearman rank correlation (marked correlations are significant at *p* < 0.01); included: primary and secondary school; excluded: polytechnic school and college.

CDAS	Croatia		Macedonia		BiH		Montenegro		Slovenia		Serbia		All	
	DA (P)	DA (C)	DA (P)	DA (C)	DA (P)	DA (C)	DA (P)	DA (C)	DA (P)	DA (C)	DA (P)	DA (P)	DA (P)	DA (C)
Parent’s DA	1.000000	0.408215 *	1.000000	0.609771 *	1.000000	0.688773 *	1.000000	0.282711	1.000000	0.524459 *	1.000000	0.369036 *	1.000000	0.483519 *
Child’s DA	0.408215 *	1.000000	0.609771 *	1.000000	0.688773 *	1.000000	0.282711	1.000000	0.524459 *	1.000000	0.369036 *	1.000000	0.483519 *	1.000000

DA—dental anxiety; P—parent; C—child; CDAS—Corah dental anxiety scale; BiH—Bosnia and Herzegovina; * statistical significance (*p* < 0.01)

## Data Availability

Not applicable.

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
