# Peer review of "Relationship between Children’s and Parents’ Dental Anxiety: A Cross-Sectional Study on the Six European Countries"

_dentistry, 2022, doi:10.3390/dj10110209_

Round 1

Reviewer 1 Report

- It would be great if you could provide CDAS questionnaire as a supplemental document for the readers. 

- Sometimes cultural differences determine the way anxiety is expressed. Is that accounted for in this study?

Author Response

It would be great if you could provide CDAS questionnaire as a supplemental document
for the readers.
-Thank You! We added the CDAS questionnaire as a supplemental document in Appendix 1.
Sometimes cultural differences determine the way anxiety is expressed. Is that accounted for in this
study?
- Thank You! We added an information about cultural differences which has been accounted for in this
study as all six examined countries share same/similar cultural origin.

Reviewer 2 Report

First of all congratulations to the authors for a great job and for coming up with an interesting and well-presented paper.

1. I would like to see more attention and a greater number of measures to reduce parental anxiety in the article as well. That is, how can we avoid those traumatic experiences that adults have suffered and then passed on that anxiety to their children?

2. I would like you to put a paragraph explaining the ways in which that anxiety can be transferred to children, apart from mentioning the dentist in front of them and parents taking them to the dentist. How do you feel about the lack of oral health awareness in homes and families? Perhaps you could talk about promoting oral health workshops, talks, classes in schools, etc. Further explanation of Zubna putovnica would be an example of what I mean. 

3. There are some small mistakes to correct: modeling = modelling; avoid wordy sentences, it is the duty of the parents to = the parents must or the parents have to; there are patients who have = some patients have, etc.

Without anything else to add, congratulations on your great work. 

Author Response

I would like to see more attention and a greater number of measures to reduce parental anxiety in the article as well. That is, how can we avoid those traumatic experiences that adults have suffered and then passed on that anxiety to their children?
- Thank You! We added some additional information about possible measures and how to avoid traumatic experiences that adults have suffered and later passed on that anxiety to their children. I would like you to put a paragraph explaining the ways in which that anxiety can be transferred to children, apart from mentioning the dentist in front of them and parents taking them to the dentist.
How do you feel about the lack of oral health awareness in homes and families? Perhaps you could
talk about promoting oral health workshops, talks, classes in schools, etc. Further explanation of
Zubna putovnica would be an example of what I mean.
- Thank You! We share Your opinion about the importance about the lack of oral health awareness in
homes and families so we added more information about Zubna putovnica as a way to raise awareness
among parents and improve oral health in Croatia. Also, we added more information about the possible
ways of transferring anxiety to children.
There are some small mistakes to correct: modeling = modelling; avoid wordy sentences, it is the
duty of the parents to = the parents must or the parents have to; there are patients who have = some
patients have, etc.
- Thank You! We corrected this words/phrases and engaged a native speaker to lecture the writing.

Reviewer 3 Report

The manuscript titled (Relationship between children's and parents' dental anxiety) with the aim is to investigate the relationship between children's and parents' dental anxiety.

Introduction:

- "Therefore, it can be concluded that negative attitudes tend to continue."

You can't use the word conclude in the introduction. 

- At the end of the introduction, we need to read the aim of the study and/or the null hypothesis to be tested.

Results:

The results are a little complex; try to simplify them so the normal reader can understand. Try to make it understandable for a different specialty.

Discussion:

- You compared the data to the DMFT index" in Macedonia DMFT index was 3.5 for 12-year-olds and in Slovenia, DMFT index was 1.5 [22,23]." the two references are old compared to the same samples you collected for Macedonia and Slovenia. Please revise or delete.

-"A parent-child anxiety correlation of 0.4 in our study suggests a strong association" would you explain the correlation number (0.4) in which table? and try to be consistent by using commas or periods.

- "We explain this with the successfully introduced national program in the Republic of Croatia, which has been implemented" Please don't advertise your previous work unless it is a published paper and you refer to it.

 Conclusions:

The conclusion is long and contains the word "in conclusion" at the end, making it looks like a conclusion for the conclusion.

Author Response

About the Title of the article, I suggest you to modify it and add the type of article.
- Thank You! We modified the title of the article and added an information about the type of article.
The introduction section is very short and is needed to add other references to increase the quality of
the manuscript, add recent references about the topic of the article, dwelling in the introduction on
articles published in 2022 and describing what your article will add compared to the last articles
published; Preferably a published articles should be with 90 or more references.
- Thank You! We added some recent references to improve the introduction section as well as some
additional text in this part of the manuscript.
I suggest you some articles (about Psychological aspect and Oral diseases, and about Health
educations today performed also trough telemedicine that is often useful for parents.) that will help
you improve your article.
- Thank You for Your suggestions! We included all three suggested articles in the manuscript as they helped
us to improve quality of the manuscript.
You need to review the grammar and English of your article.
- Thank You! We engaged a native speaker to lecture the writing.
I suggest you to add an image in order to improve the iconography of the article.
- Thank You! Due to Your suggestion, we decided to change data shown in Table 1. into Figure 1.
Please expand conclusion section with main results and future perspectives of this study
- Thank You! We expanded this part of the manuscript with the main result of the study and commented
what are the future perspectives of this study.

Reviewer 4 Report

Dear Authors the paper ” Relationship between children's and parents' dental anxiety‘’is really interesting, well conducted and fits the objectives of the journal; but it is necessary to review some points in order to improve the quality of the paper: 

-About the Title of the article, I suggest you to modify it and add the type of article.

- The introduction section is very short and is needed to add other references to increase the quality of the manuscript, add recent references about the topic of the article, dwelling in the introduction on articles published in 2022 and describing what your article will add compared to the last articles published; Preferably a published articles should be with 90 or more references.

I suggest you some articles (about Psychological aspect and Oral deseases, and about Health educations today performed also trough telemedicine that is often useful for parents.) that will help you improve your article.

Psychiatric disorders in oral lichen planus: A preliminary case control study PubMed ID 29460524 

How social media meet patients’ questions: YouTubE™ review for children oral thrush PubMed ID 29460525 
Teledentistry in the Management of Patients with Dental and Temporomandibular Disorders Doi: https://doi.org/10.1155/2022/7091153 

-You need to review the grammar and English of your article.

-I suggest you to add an image in order to improve the iconography of the article.

-Please expand conclusion section with main results and future perspectives of this study

Kind Regards

Author Response

About the Title of the article, I suggest you to modify it and add the type of article.
- Thank You! We modified the title of the article and added an information about the type of article.
The introduction section is very short and is needed to add other references to increase the quality of
the manuscript, add recent references about the topic of the article, dwelling in the introduction on
articles published in 2022 and describing what your article will add compared to the last articles
published; Preferably a published articles should be with 90 or more references.
- Thank You! We added some recent references to improve the introduction section as well as some
additional text in this part of the manuscript.
I suggest you some articles (about Psychological aspect and Oral diseases, and about Health
educations today performed also trough telemedicine that is often useful for parents.) that will help
you improve your article.
- Thank You for Your suggestions! We included all three suggested articles in the manuscript as they helped
us to improve quality of the manuscript.
You need to review the grammar and English of your article.
- Thank You! We engaged a native speaker to lecture the writing.
I suggest you to add an image in order to improve the iconography of the article.
- Thank You! Due to Your suggestion, we decided to change data shown in Table 1. into Figure 1.
Please expand conclusion section with main results and future perspectives of this study
- Thank You! We expanded this part of the manuscript with the main result of the study and commented
what are the future perspectives of this study.

We look forward to hearing from You in due time regarding our submission and to respond to any further
questions and comments You may have.
Best regards,
Authors
